# Image-free Pre-training for Low-Level Vision

## ABSTRACT

The constrained data scale in low-level vision often induces the demon overfitting hazard for restoration networks, necessitating the adoption of the pre-training paradigm. Mirroring the success of the high-level pre-training approaches, recent methods in the low-level community aim to derive general visual representation from extensive data with synthesized degradation. In this paper, we propose a new perspective beyond the data-driven image pre-training paradigm for low-level vision, building upon the following examination. First, unlike the semantic extraction prevalent in high-level vision tasks, low-level vision primarily focuses on the continuous and content-agnostic pixel-level regression, indicating that the diversified image contents inherent in large-scale data are potentially unnecessary for low-level vision pre-training. Secondary, considering the low-level degradations are highly relevant to the frequency spectrum, we discern that the low-level pre-training paradigm can be implemented in the Fourier space with fostered degradation sensibility. Therefore, we develop an Image-Free Pre-training (IFP) paradigm, a novel low-level pre-training approach with necessity of single randomly sampled Gaussian noise image, streamlining complicated data collection and synthesis procedure. The principle of the IFP involves reconstructing the original Gaussian noise from the randomly perturbed counterpart with partially masked spectrum band, facilitating the capability for robust spectrum representation extraction in response to the capricious downstream degradations. Extensive experiments demonstrate the significant improvements brought by the IFP paradigm to various downstream tasks, such as 1.31dB performance boost in low-light enhancement for Restormer, and improvements of 1.2dB in deblurring and 2.42dB in deraining for Uformer.

## CCS CONCEPTS

• **Computing methodologies → Artificial intelligence**; **Computer vision**; *Computer vision representations*.

## KEYWORDS

Low-level Pre-training, Fourier Transform, Masked Image Modeling

## 1 INTRODUCTION

In order to alleviate the overfitting problem and bring further improvement when data is limited, image pre-training has received widespread attention in the field of computer vision. However, unlike the remarkable success in high-level vision, the application of

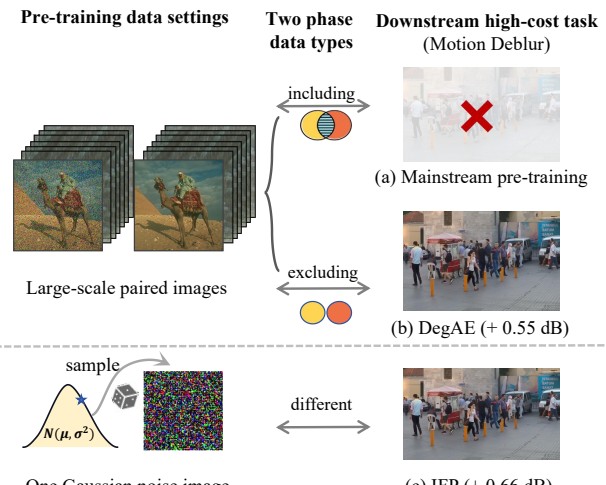

Figure 1: Comparison of different low-level vision pre-training methods. (a) The data types utilized in both the pre-training and fine-tuning phases of mainstream methods are identical. However, the necessity for large-scale paired data poses a challenge, particularly for high-cost tasks where synthesizing degraded data is challenging. (b) DegAE [29], while not requiring identical data types as downstream tasks, still mandates learning from extensive synthesized datasets. (c) In contrast, our IFP necessitates only a randomly generated Gaussian noise sample throughout the entire pre-training process, thereby offering superior adaptability and efficiency.

pre-training in low-level vision remains underdeveloped. In mainstream opinion, this is attributed to the lack of large-scale low-level vision datasets comparable to ImageNet [8] in the community. Adhering to the perspective that the scale of data plays a decisive role in the effectiveness of pre-training, pioneer works [4, 7, 23] attempt to synthesize paired degradation data on ImageNet [8] through sundry data corruption pipelines and perform corresponding image restoration tasks. These data-driven pre-training methods meet the challenge of introducing target degradation into the pre-training phase. This requirement hinders their applicability to high-cost tasks where the synthesis of degradation is a challenge.

Recently, the emergence of DegAE [28] makes it possible for high-cost tasks to benefit from pre-training. It no longer focuses on data acquisition but designs a new pre-training paradigm that learns a general low-level vision representation by transferring the degradation between images. The new paradigm significantly improves the downstream image restoration tasks after fine-tuning, even if the target degradation does not appear in the pre-training stage. However, DegAE [28] does not deviate from the traditional pre-training paradigm of learning from data. Because it still relies on

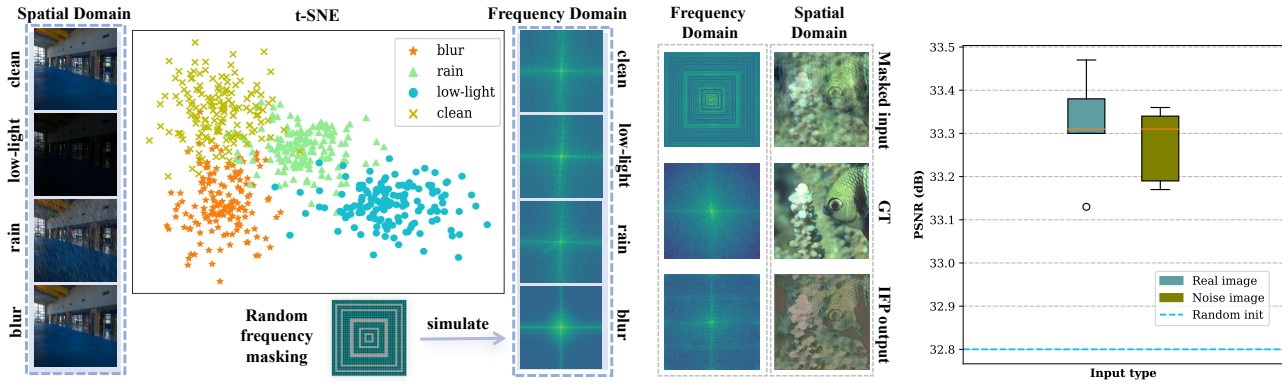

(a) Degradations in spatial domain and frequency domain

(b) Reconstruction after IFP

(c) The effect of different image types adopted in IFP

**Figure 2: (a) Both visualization pattern and t-SNE of the amplitude spectrum manifest that different degradation types exhibit disparate disturbance patterns to the frequency spectrum. We simulate different frequency disturbances by random frequency masking in IFP. (b) We perform the Frequency-aware Masking (FAM) on the real image and then reconstruct it using Restormer [45], which is only pre-trained with IFP. And all IFP needs is a randomly generated Gaussian noise image. (c) The pre-training effect of different image types adopted in IFP is similar. Random Init denotes the baseline without IFP. Real images are randomly selected from ImageNet [8]. More details are in Table 5.**

manually generating degradation data pairs exceeding 10G. Moreover, it requires an additional degradation information injection module, which necessitates significant computational resources and may potentially cause task bias in downstream applications.

Different from previous data-driven works, we propose a data-efficient and time-saving low-level pre-training paradigm called Image-Free Pre-training (IFP). As shown in Fig. 1, IFP requires only one randomly generated Gaussian noise image. However, it can enhance the performance of Restormer [45] on motion deblurring task by 0.66 dB. Specifically, IFP performs Masked Image Modeling (MIM) [14, 37, 41] in the frequency domain, enabling the model to deduce the information of the masked bands from the known bands. Such design is based on the following observations:

First, we find that different degradation types show different disturbance patterns to the spectrum in the frequency domain. As illustrated in Fig. 2(a), blurring predominantly eliminates high-frequency details, rain primarily introduces low-frequency information, and low-light augments low-frequency while diminishing high-frequency. Therefore, image restoration can be interpreted as the process of reconstructing any destroyed frequency band by using the unchanged ones. Frequency domain MIM enables models to learn occlusion invariance on the spectrum, thereby making it suitable as the pre-training task for low-level vision.

Second, unlike semantic extraction in high-level vision, the core intention in low-level pre-training lies in the continuous pixel-level regression ability, which is principally content-agnostic. Inspired by the generative models that takes Gaussian noise to synthesis various content images, we take the Gaussian noise as input and simulate different frequency disturbances by random frequency masking to investigate content-agnostic pre-training. It can be observed from Fig. 2(b) that albeit the model has only seen a randomly generated Gaussian noise image during pre-training, it still gets

a strong image reconstruction ability after frequency MIM pre-training. Moreover, we show the influence of image types on the IFP effect in Fig.2(c). With applying our method on real images from ImageNet [8] dataset, the performance increase is consistent compared with random noise. These findings imply that the effectiveness of IFP does not stem from the image and the diversity of image content inherent in large-scale data is not essential.

In conclusion, IFP is a new attempt in the field of low-level vision pre-training. To our knowledge, it is the first method that deviates from the traditional pre-training paradigm of learning from data. It means that IFP is independent on downstream tasks, and the pre-trained representations are general and transferable. Moreover, only one Gaussian noise image means the computational cost comes to be extremely low. In summary, our contributions include:

- We propose a simple but effective low-level pre-training paradigm, named Image-Free pre-training (IFP), where one randomly generated Gaussian noise image is all IFP needs in contrast with prevailing image pre-training methods. To the best of our knowledge, IFP is the first attempt to find a solution in frequency domain and exploit task-agnostic data for low-level vision pre-training, releasing the great potential under data-constrained conditions.
- Inspired by the remarkably disparate disturbance pattern of diverse degradations on the spectrum domain, we propose the frequency-aware masking strategy to enable the degradation-agnostic general representation learning by reinforcing the capability of the model for perceiving the susceptible spectra variations for downstream tasks.
- Extensive experiments on a variety of downstream image restoration tasks demonstrate the efficiency and effectiveness of our method, including image deraining, image deblurring, and low-light image enhancement.

**Figure 3: Overview of our IFP pre-training pipeline. The only pure noise input $I_o$ required for pre-training is randomly sampled from the Gaussian distribution $I(x) \sim \mathcal{N}(\mu, \delta^2)$ at the beginning of stage 1. During pre-training, $I_o$ is converted to the frequency domain by FFT to obtain its amplitude spectrum $A_o$ and phase spectrum $P_o$. The Frequency-aware Masking (FAM) strategy simulate random degradation effect on different frequencies. Specifically, the amplitude spectrum is randomly masked and the phase spectrum is kept unchanged. After iFFT, the noise whose amplitude information has been corrupted is sent to the encoder (e.g., ViT, CNN) and the original image is reconstructed using a lightweight decoder. At the downstream task fine-tuning stage, we initialize the model with encoder parameters obtained during the pre-training stage, replace the decoder with a simple convolutional layer, and then fine-tune the whole network with data from a specific low-level task.**

## 2 RELATED WORK

### 2.1 Image Restoration

The purpose of image restoration is to reconstruct high quality natural images from the observed degraded images (e.g. noise, blur, rain drops) by removing degradations. Early methods typically focuses on incorporating various natural image priors along with hand-crafted features for specific degradation removal tasks [1, 30]. Recently, deep learning based methods have made compelling progress on various image restoration tasks. For instance, SRCNN [9] introduces an end-to-end convolutional neural network for super-resolution task. Zhang et al. [50] proposes the first deep learning method DnCNN for denoising task, DehazeNet [3] and MSCNN [34] for image dehazing task, DeblurGAN [20] and Deblurgan-v2 [21] for image motion deblurring task. In addition, with the advent of ViT [10], the transformer, due to its excellent performance in modeling global dependencies and superior adaptability to the input content, has been introduced into vision tasks. For instance, IPT [4], Uformer [36], SwinIR [26] and Restormer [45] are notable

examples of such methods that have significantly contributed to advancements in the image restoration area.

### 2.2 Low-level Vision Pre-training

Pre-training can help the model find a good initialization, which is an effective way to alleviate model overfitting problem in data-scarce scenarios, but there is little exploration in the low-level vision field. Most low-level vision pre-training methods [4, 7, 23] concentrate on tasks where downstream training data is easy to obtain. The pre-training phase can be supplanted by adding downstream training data, thereby undermining the true purpose of pre-training. Based on the challenge of procuring training data for downstream tasks, DegAE [28] divides low-level tasks into low-cost and high-cost categories. Focusing on only high-cost tasks, DegAE [28] designs a pretext task centered around image degradation transfer, taking into account the specific characteristics of low-level tasks. The downstream tasks are significantly improved with relatively less pre-training data. However, it still requires over 10G of data that has undergone a series of manual preprocessing,

along with additional structural modules. In contrast, our proposed IFP does not require any preprocessing, and the data requirement is only a randomly generated Gaussian noise image, which means the degradation invariance is obtained by design rather than data.

## 3 METHOD

The overall architecture of IFP is shown in Fig. 3, which uses the target model as the encoder and a simple CNN as the decoder. One randomly generated Gaussian noise image with a random amplitude spectrum mask is used as input to perform image reconstruction, and the lost amplitude spectrum information is attempted to be restored from the generated image. Despite performing masking in the frequency domain, we still take the converted spatial images as input such that the model would not suffer an input domain gap between pre-training and fine-tuning. In the forthcoming sections, we will elaborate on how to generate the noise input in Sec. 3.1 and how to carry out frequency-aware masking (FAM) strategy in Sec. 3.2. In addition, The design of the encoder and decoder in the framework is introduced in Sec. 3.3 and the optimization objective of the pre-training process is proposed in Sec. 3.4. Finally, we discuss the essence of IFP in Sec. 3.5.

### 3.1 Noisy Input

Previous research [19] has shown that the benefit from MIM is free of content information in the spatial domain. In Sec. 4.5, we confirm that IFP similarly exhibits this characteristic in the frequency domain. Therefore, for the input, we select Gaussian noise $I(x) \sim \mathcal{N}(\mu, \delta^2)$ with $\mu = 0$ and $\delta = 1$ as standard to simplify natural image representation. The theoretical foundation of this operation is rooted in the fact that its statistical property aligns with the Gray-World Color Constancy Hypothesis [2], which posits that for an image with large color variations, the averages of the three RGB components converge to the same gray value K. In the pre-training stage, we only initialize a Gaussian noise image randomly at the beginning of the process.

### 3.2 Frequency-aware Masking

Our Frequency-aware Masking (FAM) is performed on the amplitude spectrum for the following reasons: (1) The effect of degradation on the original image appears as the random destruction of low or high frequency information in Fourier space. (2) Phase represents the content information and spatial details while amplitude represents the global statistical properties. We want to avoid the influence of specific image content when designing the pretext task. The FAM operation is simple. Specifically, since the destruction of degradation on the frequency spectrum is center-symmetric and discrete, we don't divide the input image into patches like MAE [14], nor do we directly use a high-pass or low-pass filter to mask the continuous frequency band like [40], but regarding the amplitude spectrum of the image as a combination of concentric squares with unit width. Then, following MAE [14], we randomly sample these concentric squares at a 25% ratio and follow a uniform distribution. Algorithm 1 provides the pseudo-code to show the details of FAM. Such operation can largely eliminate the information redundancy caused by the continuity of frequency distribution.

---

**Algorithm 1** Pseudocode of FAM operation in IFP.

```
# x : data input (WxHxC)
# m_ratio: mask ratio

# calculate the mask range
mid_x = W/2
mid_y = H/2
max_r = min(mid_x, mid_y)

# calculate the number of mask frames
mask_num = max_r * m_ratio
# random generate mask frames
mask_ls = randint(0, max_r, mask_num)
mask = ones(W, H) # WxH
for i in mask_ls: # generate mask
    mask[mid_y-i, mid_x-i : mid_x+i] = 0
    mask[mid_y+i, mid_x-i : mid_x+i] = 0
    mask[mid_y-i : mid_y+i, mid_x-i] = 0
    mask[mid_y-i : mid_y+i, mid_x+i] = 0

# masking
x_m, x_p = fft(x) # magnitude, phase
x_m_masked = x_m * mask # random mask

# 2D iFFT
x_corrupted = ifft(x_m_masked, x_p)

return x_corrupted
```

---

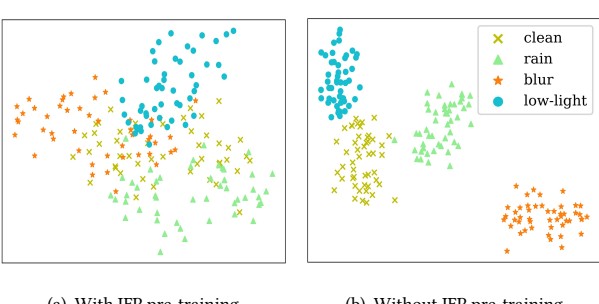

(a) With IFP pre-training  (b) Without IFP pre-training

**Figure 4: Visualization of degradation embeddings from Restormer [45]. With IFP pre-training, the embeddings for each task are closer to the clean ones.**

### 3.3 Encoder and Decoder

We use the model that needs to be pre-trained as the encoder of the IFP model during pre-training phase, which does not have a fixed form. Since there is no single architecture that works best on all low-level vision tasks, in this paper we employ two state-of-the-art Transformer architectures in low-level vision - Ufomer [36] and Restormer [45] as our encoder for multiple tasks. For the model that we use as an encoder, nothing else is done except to modify the channel number of the last convolutional layer from 3 to 64 to accommodate subsequent decoder. This operation ensures that the original architecture of the model is largely preserved, allowing it to retain its inherent strengths while benefiting from the additional learning provided by the pre-training process. Since the decoder is only used in the pre-training stage to perform image reconstruction tasks, its architecture can be flexibly designed in a way that is independent of the encoder design. As long as its input is compatible

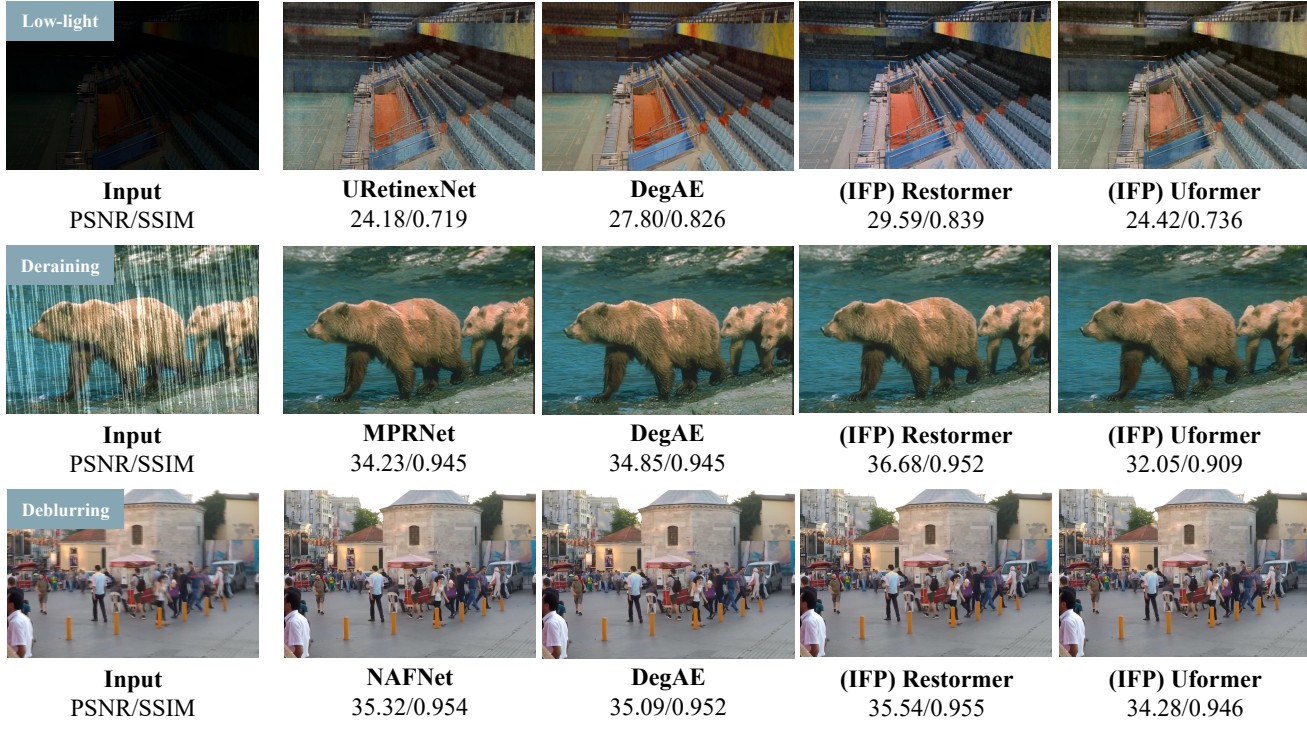

| | Input | URetinexNet | DegAE | (IFP) Restormer | (IFP) Uformer |
|---|---|---|---|---|---|
| **Low-light** | PSNR/SSIM | 24.18/0.719 | 27.80/0.826 | 29.59/0.839 | 24.42/0.736 |
| **Deraining** | Input PSNR/SSIM | MPRNet 34.23/0.945 | DegAE 34.85/0.945 | (IFP) Restormer 36.68/0.952 | (IFP) Uformer 32.05/0.909 |
| **Deblurring** | Input PSNR/SSIM | NAFNet 35.32/0.954 | DegAE 35.09/0.952 | (IFP) Restormer 35.54/0.955 | (IFP) Uformer 34.28/0.946 |

**Figure 5: Visual comparison with state-of-the-art methods on three low-level vision tasks. Please zoom in for details.**

with the output of the encoder and simple enough, it can be in any form. Follow DegAE [28], we simply take a CNN with four residual blocks [15] as our decoder and it will be replaced by a single convolution layer in the downstream task fine-tuning stage.

## 3.4 Optimization Objective

Our IFP reconstructs the input by predicting the missing values on the amplitude spectrum. Specifically, the loss function consists of amplitude spectrum reconstruction loss and pixel-wise reconstruction loss, which can be formulated as:

$$\mathcal{L} = \lambda \|I_o, I_r\|^2 + \left\|\overline{M} \odot A_o, \overline{M} \odot A_r\right\|^2, \tag{1}$$

where $I_o$ denotes the original Gaussian noise image, $I_r$ is the reconstructed image, $\lambda$ is the balanced weight. $\overline{M}$ represents the binary masking map, $\odot$ is the Hadamard product between matrices, $A_o$ and $A_r$ are the amplitude spectra corresponding to $I_o$ and $I_r$ respectively. The first part of the formula is the pixel-wise reconstruction loss, while the second part is the amplitude spectrum reconstruction loss. We follow the settings in MAE [14], and only calculate the masked region in practical application.

## 3.5 Discussion About How IFP Works

IFP is the first attempt to exploit task-agnostic data for low-level vision pre-training. We argue that by learning occlusion invariance in the frequency domain, IFP enables the model to extract robust spectral representation regardless of whether it encounters a clean or degraded image. To substantiate this claim, we take Restormer [45]

as the backbone and perform the visual analysis of its output features before and after pre-training. As shown in Fig. 4, with the help of IFP pretraining, the gap between different degradations is reduced, proving advantageous in response to the unpredictable disturbances caused by downstream degradations.

## 4 EXPERIMENT

We conduct extensive experiments to show the effectiveness of our proposed IFP method. Specifically, We evaluate the proposed IFP pre-training paradigm on several high-cost low-level tasks, including image deraining, image deblurring, and low-light image enhancement. In the following sections, we explain the implementation details and show the IFP's performance on different tasks. Due to the space limit, we show the experimental results for the low-cost task in the supplementary file.

## 4.1 Implementation Details

We implement IFP on a single NVIDIA Geforce RTX 3090 GPU. For pre-training, the learning rate is initialized as 2e-4. Adam optimizer [18] with $\beta_1 = 0.9$ and $\beta_2 = 0.99$ is adopted. The Gaussian noise input selected from $I(x) \sim \mathcal{N}(0, 1)$ is in the shape of 224×224. The batch size is one and a total of 30K iterations are executed. After pre-training, we initialize the model with the parameters of the encoder from the first stage and then fine-tune it on specific downstream datasets. We employ the same training policy across different backbones for fairness and convenience. It's important to

| Restormer | Restormer + IFP | Uformer | Uformer + IFP |

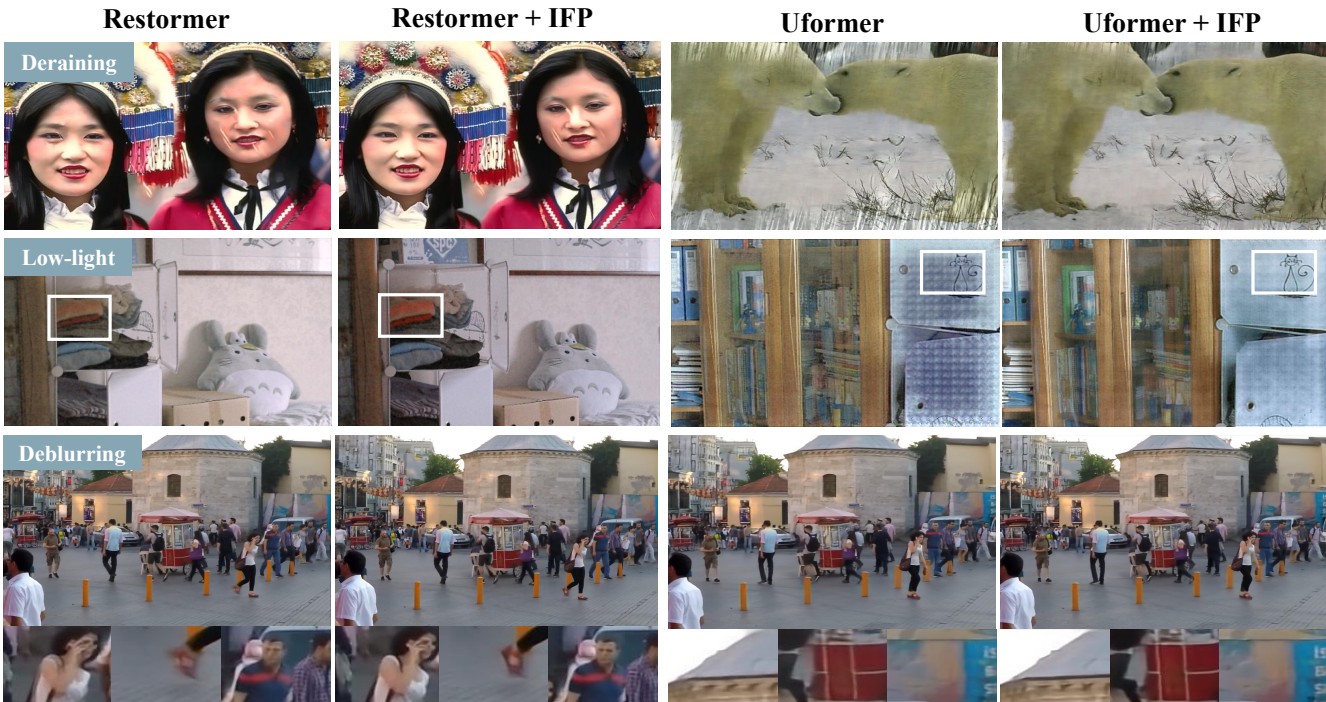

**Figure 6: Visual comparisons on deraining, deblurring and low-light image enhancement tasks. Please zoom in for details.**

highlight that since our IFP pre-training data is only one noise image, the results shown below are the average of 5 different Gaussian noise results to ensure the reliability of the experimental results.

## 4.2 Deraining

We adopt Rain13K dataset for fine-tuning, which includes 13,712 clean-rain image pairs collected from multiple datasets [11, 24, 25, 29, 44] and is newly-adopted in [6, 36, 45, 46]. For testing, we adopt Rain100L [43], Rain100H [43], Test100 [49], Test1200 [48] and Test2800 [12] datasets. The PSNR and SSIM values is calculated on the Y channel in the YCbCr color space. We compare our IFP with state-of-the-art methods, including five derain methods: DerainNet [11], RESCAN [24],PreNet [33], MSPFN [16],MPRNet [46], and one state-of-the-art pre-training method: DegAE [28]. The quantitative results are shown in Table 1 and the visual results are shown in Fig. 5. It shows that IFP helps improve the performance of Restormer and Uformer on all five datasets. The improvement is the most obvious on Rain100H dataset, with Uformer producing an improvement of 2.42dB and Restormer producing a performance improvement of 0.84dB. From the visualization results in Fig. 6, IFP pre-training can help the target model to remove rain more thoroughly to produce better visual effects.

## 4.3 Deblurring

For image deblurring, we adopt GoPro [31] dataset for training and testing. It consists of 3214 pairs of blurred and clean images extracted from 33 sequences. The blurred images are generated by averaging varying number (7−13) of successive latent frames

to produce varied blur. Follow previous methods [31, 47], we use 2103 image pairs for training and the remaining 1111 pairs for testing. the compared methods that we select are DeblurGAN [20], DeblurGAN-v2 [21], SRN [35], SPAIR [32], HINet [6], MPRNet [46], IPT [4], NAFNet [5] and DegAE [28]. As can be seen from the quantitative results of motion deblur in Table 2, IFP has brought improvements of 0.60 dB and 1.02 dB to Restormer and Uformer on GoPro dataset, respectively. Although neither Restormer nor Uformer achieves the best performance on the GoPro dataset, it does not affect the validation of our method. One can achieve better results by using a higher-performance backbone. Moreover, as illustrated in Fig. 6, with application of IFP pre-training, the model becomes more proficient in eliminating blur from the image and the image restored is closer to the target with more details.

## 4.4 Low-light Image Enhancement

IFP can also bring considerable improvement on low-light enhancement task. We use the LOL dataset [38] as the fine-tuning dataset, which contains 500 low/normal-light image pairs. In the training, we merely employ 450 image pairs, and no synthetic images are used. We report the results of IFP along with eight state-of-the-art methods TAPE [27], AirNet [22], PairLIE [13], EnlightenGAN [17], STAR [42], URetinexNet [39], SwinIR [26] and DegAE [28]. As can be seen from the results in Table 3 and Fig. 5, with the introduction of IFP, the performance of the model has been significantly improved. Specifically, Uformer and Restormer achieve 0.94 dB and 1.31dB improvements on the LOL dataset, respectively. What's

**Table 1: Image deraining results on benchmark datasets. IFP pre-training can significantly boost the performance of the model, such as the improvement on Uformer backbone brings up to 2.42dB.**

| Method | Rain100L | | Rain100H | | Test100 | | Test1200 | | Test2800 | |
| --- | --- | --- | --- | --- | --- | --- | --- | --- | --- | --- |
| | PSNR↑ | SSIM↑ | PSNR↑ | SSIM↑ | PSNR↑ | SSIM↑ | PSNR↑ | SSIM↑ | PSNR | SSIM↑ |
| DerainNet [11] | 27.03 | 0.884 | 14.92 | 0.592 | 22.77 | 0.810 | 23.38 | 0.835 | 24.31 | 0.861 |
| RESCAN [24] | 29.80 | 0.881 | 26.36 | 0.786 | 25.00 | 0.835 | 30.51 | 0.882 | 31.29 | 0.904 |
| PreNet [33] | 32.44 | 0.950 | 26.77 | 0.858 | 24.81 | 0.851 | 31.36 | 0.911 | 31.75 | 0.916 |
| MSPFN [16] | 32.40 | 0.933 | 28.66 | 0.860 | 27.50 | 0.876 | 32.39 | 0.916 | 32.82 | 0.930 |
| MPRNet [46] | 36.40 | 0.965 | 30.41 | 0.890 | 30.27 | 0.897 | 32.91 | 0.916 | 33.64 | 0.938 |
| DegAE(Restormer) [28] | 37.85 | 0.973 | 30.72 | 0.896 | 29.85 | 0.899 | 30.84 | 0.877 | 33.88 | 0.939 |
| Uformer | 32.56 | 0.943 | 21.81 | 0.751 | 25.65 | 0.864 | 31.59 | 0.915 | 29.26 | 0.913 |
| IFP(Uformer) | **32.63** | 0.941 | **24.23** | **0.805** | **26.24** | **0.872** | **31.59** | 0.913 | **29.27** | **0.913** |
| Restormer | 38.22 | 0.975 | 30.59 | 0.894 | 30.01 | 0.904 | 31.33 | 0.904 | 33.84 | 0.942 |
| IFP(Restormer) | **38.68** | **0.977** | **31.43** | **0.902** | **30.32** | **0.916** | **32.05** | **0.904** | **33.86** | **0.944** |

**Table 2: Image deblurring results on Gopro dataset. IFP pre-training can bring improvement up to 1.02dB for Uformer backbone.**

| Method | GoPro | |
| --- | --- | --- |
| | PSNR↑ | SSIM↑ |
| DeblurGAN [20] | 28.70 | 0.927 |
| DeblurGAN-v2 [21] | 29.55 | 0.934 |
| SRN [35] | 30.26 | 0.934 |
| SPAIR [32] | 32.06 | 0.953 |
| MPRNet [46] | 32.66 | 0.959 |
| DegAE(Restormer) [28] | 32.67 | 0.928 |
| HINet [6] | 32.71 | 0.959 |
| NAFNet [5] | 32.85 | 0.960 |
| Uformer | 30.97 | 0.903 |
| IFP(Uformer) | **31.99(+1.02)** | **0.919** |
| Restormer | 32.12 | 0.926 |
| IFP(Restormer) | **32.72(+0.60)** | **0.929** |

**Table 3: Quantitative comparisons on low-light image enhancement dataset. IFP pre-training can bring improvement up to 1.31dB for Restormer backbone.**

| Method | LOL | |
| --- | --- | --- |
| | PSNR↑ | SSIM↑ |
| EnlightenGAN [17] | 17.48 | 0.651 |
| SwinIR [26] | 17.81 | 0.723 |
| AirNet [22] | 18.18 | 0.735 |
| TAPE [27] | 18.97 | 0.621 |
| STAR [42] | 19.30 | 0.579 |
| PairLIE [13] | 19.51 | 0.736 |
| URetinexNet [39] | 19.84 | 0.826 |
| DegAE(Restormer) [28] | 23.37 | 0.874 |
| Uformer | 19.63 | 0.737 |
| IFP(Uformer) | **20.57(+0.94)** | **0.755** |
| Restormer | 22.40 | 0.873 |
| IFP(Restormer) | **23.71(+1.31)** | **0.876** |

more, as demonstrated in Fig. 6, after IFP pre-training, the model exhibits enhanced capabilities in restoring darker regions of the image, and the problems of in-homogeneous background and abnormal color have been significantly improved.

## 4.5 Ablation Study

We present the ablation experiments on image deraining task with Restormer [45] as the backbone to verify the effectiveness of our method. Basically, we explore the impact of pre-training iterations, the number of Gaussian noise images, and the masking ratio on the proposed IFP method. Additionally, to verify the data-agnostic characteristic of the IFP pre-training method, we conducted comparative experiments using different real images and Gaussian noise images. We still train on the Rain13K dataset [11, 24, 25, 29, 44] and test on five benchmark datasets: Rain100L [43], Rain100H [43],

Test100 [49], Test1200 [48] and Test2800 [12]. The average performance across these test datasets is used as the evaluation criterion for the models' performance.

**Type of pre-training input.** In order to verify whether the performance of the IFP pre-training method is affected by the input image content, we conduct a total of 10 repeated experiments on the image deraining task using 5 randomly generated Gaussian noise images and 5 real images randomly selected from ImageNet [8]. The final results are shown in Table 5. It can be observed that the difference between pure Gaussian noise images and real images in terms of downstream fine-tuning improvement is very small. The performance of Gaussian noise images is even more stable than that of real images. This result indicates that the effectiveness of the IFP method does not stem from the content of the image. Such data-agnostic invariance indicates that IFP pre-training method is

independent of downstream low-level vision tasks, and the pre-trained representations are general and transferable. IFP unleashes significant potential under data-constrained conditions.

**Table 4: Ablation results on the pre-training iterations of IFP. When fine-tuning iterations reaches 500K, the Restormer backbone's result is 32.52 dB. And when fine-tuning iterations is 800K, the Restormer backbone's result is 32.80 dB.**

| | Finetuning iterations | Pre-training iterations | | | | |
|---|---|---|---|---|---|---|
| | | 10K | 20K | 30K | 40K | 50K |
| Mean | 500K | 32.82 | 32.86 | 32.99 | 32.86 | 32.85 |
| | 800K | 33.10 | 33.12 | 33.27 | 33.15 | 33.13 |
| Variance | 500K | 0.015 | 0.018 | 0.022 | 0.054 | 0.032 |
| | 800K | 0.010 | 0.008 | 0.009 | 0.019 | 0.010 |

**Table 5: The effect (PSNR) of different image types adopted in IFP paradigm. Random Init denotes the baseline without IFP pre-training.**

| Input type | Number | Training | Mean | Variance |
|---|---|---|---|---|
| Real | 5 | Random Init | 32.80 | - |
| | | IFP | 33.32 | 0.016 |
| Noise | 5 | Random Init | 32.80 | - |
| | | IFP | 33.27 | 0.008 |

**Table 6: Ablation experiment results on the number of the randomly generated Gaussian noise images adopted in IFP. Random Init denotes the baseline. Results in bold indicate the best performance (PSNR) as default settings.**

| Pre-training images | Fine-tuning iterations | | | |
|---|---|---|---|---|
| | 100k | 600k | 800k | 900k |
| Random Init | 30.36 | 32.75 | 32.80 | 32.80 |
| 1 | **32.03** | **33.14** | **33.27** | **33.29** |
| 10 | 30.76 | 32.77 | 32.83 | 32.96 |
| 100 | 30.68 | 32.60 | 32.74 | 32.86 |

**Number of pre-training iterations.** We explore the influence of pre-training degree on the downstream deraining task. As shown in Table 4, we are surprised to find that when the number of pre-training iterations reaches about 10K, there is a certain performance improvement. When the number of iterations reaches about 30K, the pre-training achieves the ideal effect, and the performance improvement reaches 0.47dB. When the number of iterations reaches about 40K, it can still significantly improve the model's performance, but the effect slightly decreases. We hypothesize that this is because overfitting tends to occur when performing MIM training on a very small training set, as the network can easily "remember" the unique training image.

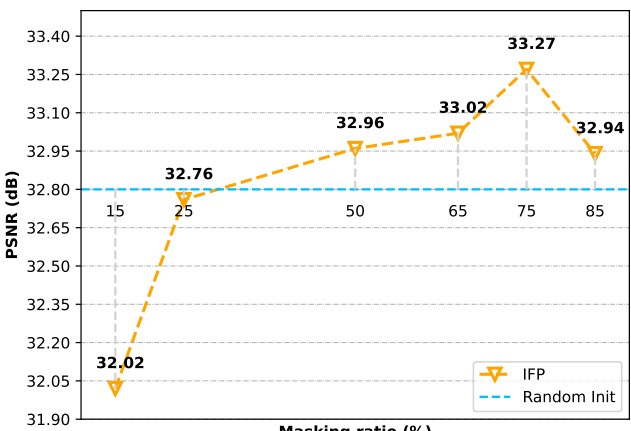

**Figure 7: Ablation experiment results on the masking ratios adopted in IFP. Random Init denotes the baseline and its result is 32.80 dB.**

**Masking ratio.** We also explore the influence of the masking ratio on IFP, and the results are shown in Fig. 7. It can be observed that the effect of IFP is closely related to the masking ratio. A relatively high masking ratio can bring a good initialization to downstream tasks. Similar to the spatial MIM method [14], the IFP method works best when the masking ratio reaches 75%.

**Number of noise images.** The outstanding advantage of our method is that only one noise image is used during pre-training. Here we explore the effect of increasing the number of training inputs on model performance, and the results are shown in Table 6. It suggests that an increase in the quantity of noise images can actually hinder the progress of pre-training. We hypothesize that this phenomenon is due to the fact that the efficacy of the IFP is not intrinsically derived from the data and one image is enough for IFP to learn. Consequently, an increase in data complexity merely introduces negative interference to the model, amplifies the training challenge, thus leads to the reduction in the effectiveness of IFP.

## 5 CONCLUSION

Due to the constrained data scale related to many degradation restoration problems and the black-box nature of low-level model learning process, pre-training has not been well developed in the low-level vision field. In this paper, we propose a bold idea when thinking about solving the problem: can pre-training be done without using any task-related data? We analyze the impact of different degradation on images from the frequency domain perspective, and then introduce a new low-level self-supervised pre-training paradigm, called Image-Free Pre-training (IFP). The final results provide many surprises, and experimental results show that even if only one randomly generated Gaussian noise image is used for IFP pre-training, it can still bring significant improvement to a wide range of downstream tasks. This groundbreaking work forces us to rethink the complexity of image features in low-level vision problems, and we hope that our work will inspire research on the interpretability of low-level vision networks.

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
