# OpenReview forum: "Image-free Pre-training for Low-Level Vision"
_acmmm.org/ACMMM/2024/Conference — MM2024 Poster_

### Official Review · Reviewer_4Qnu · 2024-05-09

**Rating:** 5
**Confidence:** 4

**Summary:**

This paper presents a novel pre-training method for low-level vision that requires no data. The proposed method utilizes FFT to transform into frequency domain and then realize pre-training by Mask-Image-Modelling. With a single noisy image sampled from Gaussian distribution, this method has achieved great competitive improvement compared to other pre-training methods like DegAE, which is hungry with a huge amount of data and requires a long time to train.
Overall, this paper implements pre-training in a very novel and efficient way. I think it is of high quality and deserves to be accepted. Open source is also expected.

**Strengths:**

The motivation for this article is very clear and convincing. Different degradations show different distributions in the frequency domain, so it is very reasonable to restore different frequency components in the frequency domain as pre-training. At the same time, the authors point out that low-level vision does not rely on semantics, so only pure noise images are needed to satisfy such pre-training, which ease the hunger for data.
The proposed method is very novel and effective. Experiments are also fully conducted, demonstrating the effectiveness of the proposed method and its advantages over DegAE: it only requires a short period of pre-training and a single noise picture.

**Limitations:**

In the ablation study Table.6, increasing the number of noise pictures affects the performance. I think the author's explanation for this is too brief and not very convincing.

**Suitability:**

3

---

### Official Review · Reviewer_pDVh · 2024-05-22

**Rating:** 4
**Confidence:** 4

**Summary:**

This paper proposes a novel paradigm for low-level vision pre-training called Image-free Pre-training (IFP). It introduces a new perspective for low-level pre-training beyond data-driven image pre-training. IFP requires only A SINGLE randomly generated Gaussian noise image for pretraining, which is really interesting. IFP applies random masking on the frequency domain on the noise image, simulating random degradations in the Fourier domain for content-agnostic representation learning. Experiments demonstrate effectiveness of IFP on various downstream tasks like deraining, deblurring and low-light enhancement, achieving performance boost compared to existing baselines.

**Strengths:**

-	The paper is well written and the motivation, method and experiments are clearly explained.
-	The core idea of IFP is novel in that it deviates from traditional data-driven pre-training and operates directly in the frequency domain using only noise, which is an interesting new perspective for low-level vision.
-	The authors conduct plenty of ablation studies and obtain lots of interesting observations.
-	The visual and quantitative results show the effectiveness of IFP.

**Limitations:**

-	In Fig. 2, how are the t-SNE results obtained? In what domain and how each point obtained from the image pixel?
-	In Fig. 2(c), what is the hollow circle in the plot?
-	How well is the reconstruction result in the Image-free masking pretraining? (reconstructed noise image)
-	In Fig. 3, the ‘random process’ is too large.
-	Explanation of how IFP enables learning degradation-invariant representations could be clearer. What exactly is learned from reconstructing the noise?
-	Impact of noise type (other than Gaussian) is worth exploring.

**Suitability:**

2

---

### Official Review · Reviewer_FcJc · 2024-05-23

**Rating:** 5
**Confidence:** 3

**Summary:**

This paper introduces Image-Free Pre-training (IFP), a novel low-level pre-training paradigm for vision restoration tasks, utilizing a single Gaussian noise image and focusing on frequency domain manipulation for spectrum representation extraction in first stage pretraining; In second stage, the pretrained model is fine-tuned in various downstream tasks, which demonstrates significant improvements in various tasks such as deblurring, deraining, and low-light image enhancement.

**Strengths:**

The idea of the paper, i.e. using pure noise for the first stage pre-training is novel and interesting, and the experiments are adequate. The innovation of the paper comes mainly from the use of pure noise to train the model in the first stage, which in my understanding is similar to allowing the model to learn a better initial distribution of parameters, making it easier to train downstream tasks closer to the global optimum. This process has a very small cost in terms of data and time and can bring significant performance improvements, which I think is valuable for the community.

**Limitations:**

1.	The blurred images shown in the experimental part of the paper are still sharp, which does not reflect the effect of deblurring. Some deblurring results with high-degree blur need to be shown, e.g., those blurred images where the content of the image is no longer readable.
2.	Some of the tasks in the paper use datasets that are purely synthetic, such as GoPro in the deblurring task. Thus, some experiments on real-world data should be conducted, which is important to prove the effectiveness in reality of the proposed method.
3.	The proposed method is two-stage, thus the experimental results of the one-stage should be ablated, i.e: for each task, the input data could consist of two parts (task data and noise) for one-stage training. This experiment helps us to understand whether it is really necessary to choose a two-stage process, as one-stage tends to be more convenient and performs better.

**Suitability:**

3

---

### Official Review · Reviewer_PFuw · 2024-05-27

**Rating:** 3
**Confidence:** 4

**Summary:**

This paper proposes an Image-Free Pre-training (IFP) paradigm, a novel low-level pre-training approach with necessity of single randomly sampled Gaussian noise image, streamlining complicated data collection and synthesis procedure. The principle of the IFP involves reconstructing the original Gaussian noise from the randomly perturbed counterpart with partially masked spectrum band, facilitating the capability for robust spectrum representation extraction in response to the capricious downstream degradations.

**Strengths:**

1. This paper proposes a pre-training paradigm, called Image-Free pre-training (IFP).
2. This paper proposes the frequency-aware masking strategy to enable the degradation-agnostic general representation learning by reinforcing the capability of the model for perceiving the susceptible spectra variations for downstream tasks.
3. IFP is the first method that deviates from the traditional pre-training paradigm of learning from data.

**Limitations:**

1. From Table 1-3, it is obviously that the proposed IFP can improve the performance of low-level vision methods. However, how is the performance on real-world data? Could you provide qualitative and quantitative comparisons on real-world datasets? I would like to understand that the effectiveness on real data.
2. The amount of parameters and computational complexity of the proposed IFP should be discussed.
3. Could you provide more comparisons with pre-training methods?
4. From Table 1-3, the compared methods seem to be out of dated. The superiority of the proposed IFP cannot comprehensively demonstrated.
5. How to determine the optimal hyperparameters?
6. Could you provide the ablation study of the optimization function?

**Suitability:**

3

---

### Meta-Review · Area_Chair_sfL2 · 2024-07-01

**Recommendation:** Accept (Poster)
**Confidence:** 5

**Metareview:**

This paper presented a new image-free pre-training (IFP) scheme for low-level vision/image restoration tasks. The paper received review comments from four reviewers. The strengths of this work include: the new and interesting idea of image-free pre-training, and extensive experiments including ablation study to validate the main idea and claims.
The key idea of using pure noise to train models to perform downstream tasks would be a good contribution to the community and may inspire follow-up works. However, there are some limitations that could be addressed to make the paper stronger: Insufficient clarifications for some claims and statements, lack of sufficient details in some descriptions and illustrations, real-world applications, and the two-stage design. Some of them were addressed in the rebuttal, and the authors are suggested to include them in the final version of the paper. All four reviewers recommended positive scores, with three weak accept and one borderline accept.

Overall, the contribution and strengths of this paper overwhelm the limitations, and this work would be of interest to a large group of audience in ACM MM. As a result, the AC recommends Accept.